# Physiologic Mechanical Stress Directly Induces Bone Formation by Activating Glucose Transporter 1 (Glut 1) in Osteoblasts, Inducing Signaling via NAD+-Dependent Deacetylase (Sirtuin 1) and Runt-Related Transcription Factor 2 (Runx2)

**DOI:** 10.3390/ijms22169070

**Published:** 2021-08-23

**Authors:** Shu Somemura, Takanori Kumai, Kanaka Yatabe, Chizuko Sasaki, Hiroto Fujiya, Hisateru Niki, Kazuo Yudoh

**Affiliations:** 1Department of Sports Medicine, St. Marianna University School of Medicine, Sugao 2-16-1, Miyamae-ku, Kawasaki 216-8511, Japan; s3somemura@marianna-u.ac.jp (S.S.); kanaka@marianna-u.ac.jp (K.Y.); fujiya-1487@marianna-u.ac.jp (H.F.); 2Department of Orthopaedic Surgery, St. Marianna University School of Medicine, Sugao 2-16-1, Miyamae-ku, Kawasaki 216-8512, Japan; t2kumai@marianna-u.ac.jp (T.K.); h2niki@marianna-u.ac.jp (H.N.); 3Institute for Ultrastructural Morphology, St. Marianna University Graduate School of Medicine, Sugao 2-16-1, Miyamae-ku, Kawasaki 216-8512, Japan; c2sasaki@marianna-u.ac.jp; 4Department of Frontier Medicine, Institute of Medical Science, St. Marianna University School of Medicine, Sugao 2-16-1, Miyamae-ku, Kawasaki 216-8512, Japan

**Keywords:** mechanical stress, osteoblast differentiation, glucose transporter 1, sirtuin 1, runt-related transcription factor 2

## Abstract

Mechanical stress is an important factor affecting bone tissue homeostasis. We focused on the interactions among mechanical stress, glucose uptake via glucose transporter 1 (Glut1), and the cellular energy sensor sirtuin 1 (SIRT1) in osteoblast energy metabolism, since it has been recognized that SIRT1, an NAD+-dependent deacetylase, may function as a master regulator of the mechanical stress response as well as of cellular energy metabolism (glucose metabolism). In addition, it has already been demonstrated that SIRT1 regulates the activity of the osteogenic transcription factor runt-related transcription factor 2 (Runx2). The effects of mechanical loading on cellular activities and the expressions of Glut1, SIRT1, and Runx2 were evaluated in osteoblasts and chondrocytes in a 3D cell–collagen sponge construct. Compressive mechanical loading increased osteoblast activity. Mechanical loading also significantly increased the expression of Glut1, significantly decreased the expression of SIRT1, and significantly increased the expression of Runx2 in osteoblasts in comparison with non-loaded osteoblasts. Incubation with a Glut1 inhibitor blocked mechanical stress-induced changes in SIRT1 and Runx2 in osteoblasts. In contrast with osteoblasts, the expressions of Glut1, SIRT1, and Runx2 in chondrocytes were not affected by loading. Our present study indicated that mechanical stress induced the upregulation of Glut1 following the downregulation of SIRT1 and the upregulation of Runx2 in osteoblasts but not in chondrocytes. Since SIRT1 is known to negatively regulate Runx2 activity, a mechanical stress-induced downregulation of SIRT1 may lead to the upregulation of Runx2, resulting in osteoblast differentiation. Incubation with a Glut1 inhibitor the blocked mechanical stress-induced downregulation of SIRT1 following the upregulation of Runx2, suggesting that Glut1 is necessary to mediate the responses of SIRT1 and Runx2 to mechanical loading in osteoblasts.

## 1. Introduction

Osteoporosis is a common and well-known age-related disease that is characterized by low bone mass as well as bone microstructure destruction, resulting from the breakdown of bone homeostasis [1,2]. Bone homeostasis is maintained by the remodeling cycle of osteoblast-mediated bone formation and osteoclast-mediated bone resorption [3,4]. When the balance of bone homeostasis is destroyed, bone remodeling cannot maintain a constant bone mass, resulting in osteopenia and eventually osteoporosis [2,3,4,5]. 

Osteoarthritis (OA) is also a common, degenerative joint disease affecting the elderly population [6,7]. As OA progresses due to excessive mechanical loading, subchondral bones as well as articular cartilage tissues are severely degraded, leading to restricted mobility and joint malfunction and causing severe pain, resulting in major impairments in quality of life [6,8].

It is widely accepted that mechanical stress (loading) is an important factor in bone- and cartilage-associated cell differentiation and function, including those of osteocytes, osteoblasts, osteoclasts, and chondrocytes [9,10,11]. In bone biology, it has been reported that physiologic mechanical stress induces osteoblast differentiation and consequent bone formation and that this is controlled by the molecular pathways operating in osteocytes in response to mechanical stress [12,13]. It is well known that sclerostin, a mainly osteocyte product, is widely considered a key molecule in mechano-transduction, controlling the bone remodeling cycle [14,15]. This central role is accomplished through interactions between two opposing mechanisms: (1) mechanical unloading induces high levels of sclerostin, which antagonize the wingless-type (Wnt)/beta-catenin signaling pathway in osteocytes and osteoblasts, permitting simultaneous Wnt-noncanonical pathways in osteocytes and osteoclasts, directed at bone resorption; conversely, (2) mechanical loading suppresses the expression of sclerostin following the activation of Wnt/beta-catenin signaling in osteocytes, resulting in osteoblast differentiation and bone formation activity [15]. The cellular interaction of osteoblasts with osteocytes via the osteocytic sclerostin-Wnt/beta-catenin signaling pathways plays an important part in physiologic mechanical stress-mediated acceleration of osteoblast differentiation following bone formation [12,13,14,15,16]. In spite of this, it is unclear whether mechanical loading directly induces osteoblast differentiation and bone formation activity without a Wnt/beta-catenin-independent signaling mechanism connecting osteocytes and osteoblasts. 

Recently, attention has been attracted by the finding that the nicotinamide adenine dinucleotide (NAD)-dependent deacetylase sirtuin-1 (SIRT1) regulates many metabolic functions including DNA repair, genome stability, inflammatory response, apoptosis, cell cycle, mitochondrial function, cellular energy metabolism (adenosine triphosphate (ATP) production), and cellular responses to extrinsic stresses including mechanical stress [17,18,19,20,21]. It has been reported that SIRT1 promotes stress tolerance in a variety of diseases [21,22]. We also indicated that SIRT1 plays an important role in cellular energy metabolism as an energy sensor as well as an adenosine monophosphate-activated protein kinase-activated protein kinase (AMPK) in articular cartilage homeostasis and the progression of OA [23]. Several reports indicated that SIRT1 has two important roles: “regulation of cellular energy metabolism” and “stress tolerance” [21,24,25].

It has been reported that Runx2, an osteogenic transcription factor, is required for chondrocyte hypertrophy and osteogenesis (osteoblast differentiation/proliferation) in the bone and joint tissue [26,27,28]. Several studies demonstrated that the SIRT1 activator mediates the expression of Runx2 in mesenchymal stem cells [29,30] and that SIRT1 inactivation decreases the expression of Runx2 in chondrocytes [31]. These findings indicate that SIRT1 promotes osteogenesis through a mechanism involving the SIRT1–Runx2 interaction. In addition, It has been demonstrated that SIRT1 deacetylates a transcription factor, FoxOs, representing a subclass of a large family of forkhead proteins [32,33]. More recently, FoxOs (1–4) are known to control bone resorption and formation [34,35,36,37,38,39] Iyer S. et al. reported that SIRT1-mediated FoxOs deacetylation prevents this interaction and potentiates Wnt signaling, resulting in the induction of osteoblast proliferation [38,39]. They concluded that FoxOs may have a role as an osteoblast progenitor (transcription factor) and that SIRT1–FoxOs interaction participates in the osteoblast differentiation. 

Several reports have demonstrated that SIRT1 may function as a key regulator of the mechanical stress response as well as energy homeostasis [40,41,42,43]. It has been reported that the constant exposure of vascular endothelial cells to mechanical shear stress maintains vascular tone and activity, which is mediated through the SIRT1 pathway [26,28]. We postulated that the SIRT1–Runx2 pathway may function as a master regulator of osteoblast and chondrocyte activities in response to mechanical stress to bone and cartilage tissues. However, it remains unknown what kind of trigger or mechanism modulates the SIRT1–Runx2 pathway following mechanical stress-induced osteoblast and chondrocyte differentiation and bone formation.

It has been demonstrated that glucose uptake promotes osteoblast differentiation and bone formation by activating the osteogenic transcription factor, Runx2 [44,45,46,47]. Based on these findings, we speculated that the glucose transporter 1 (Glut1)–SIRT1–Runx2 pathway in osteoblasts may play some sort of role in mechanical stress-induced bone formation and osteoblast differentiation without involving the osteocytic sclerostin-Wnt/beta-catenin signaling pathway. 

## 2. Results

### 2.1. Effects of Mechanical Loading on Cellular Morphology in 3D Cell Culture

The three-dimensional (3D) cell-culture constructs described above were placed into individual wells of a 12-well culture dish and maintained in a 1.5 mL cell culture medium. Then, mechanical loading was applied to the 3D cell-culture constructs with the use of a custom-designed and -built apparatus, as shown in Figure 1A,B. Continuous physiologic compression was applied to the tested 3D constructs at 0 kPa (non-loaded) or 25.5 gf/cm^2^ (2.5 kPa).

The SEM analysis revealed that osteoblasts grew with cytoplasmic extension-denominated filopodia and with fine granules on the cell surface in the 3D cell–collagen sponge construct. After 24 h incubation, the mechanically loaded osteoblasts showed a higher expression of microgranules on the cell surface in comparison with the non-loaded osteoblasts (Figure 2A,B). The deposition of microgranules was observed on the surface of the collagen fibers around osteoblasts and the levels were higher in the mechanically loaded osteoblast groups (Figure 2B,D) than in the non-loaded osteoblast groups (Figure 2A,C).

The SEM analysis showed that chondrocytes with cell surface protrusions (cell surface glycoproteins, glycolipids, and glycans) adhered to the surface of the collagen sponge. The expressions of cell surface protrusions and microgranules were higher in the mechanically loaded chondrocytes (Figure 2F,H) than in the non-loaded chondrocytes (Figure 2E,G) in the 3D cell–collagen sponge construct.

The results clearly indicated that the mechanical force significantly induced the expression (number/cells) of cell surface microgranules in both osteoblasts (24 h incubation group: mechanical loading (−) vs. mechanical loading (+) = 24.3 ± 9.5/cells vs. 81.1 ± 30.6/cells, *p* < 0.01), (48 h-incubation group: mechanical loading (-) vs. mechanical loading (+) = 31.4 ± 10.6/cells vs. 115.8 ± 43.6/cells, *p* < 0.01) and chondrocytes (24 h-incubation group: mechanical loading (-) vs. mechanical loading (+) = 23.3 ± 9.5/cells vs. 76.4 ± 28.4/cells, *p* < 0.01), (48 h incubation group: mechanical loading (-) vs. mechanical loading (+) = 43.7 ± 14.5/cells vs. 94.1 ± 33.9/cells, *p* < 0.01) (each subgroup/time point, *n* = 12).

### 2.2. Effects of Mechanical Loading on Cellular Activities in Osteoblasts and Chondrocytes

Alkaline phosphatase (ALP) and osteocalcin are biomarkers of the early and advanced stages of osteoblast differentiation and activity, respectively. Regarding the level of ALP production from osteoblasts, the unpaired Cohen’s d between the control and the loading (+)-4 h incubation group was 0.313 (95.0% CI −0.599, 1.12) and the *p*-value of the two-sided permutation *t*-test was 0.456 (Figure 3(Aa)). The unpaired Cohen’s d between the control and the loading (+)-48 h incubation group was 1.1 (95.0% CI 0.198, 1.87), and the *p*-value of the two-sided permutation *t*-test was 0.0124 Figure 3(Ab) (each subgroup/time point, *n* = 12).

In the level of osteocalcin production, the unpaired Cohen’s d between the control and the loading (+)-48 h incubation group was 0.291 (95.0% CI −0.539, 1.2), and the *p*-value of the two-sided permutation *t*-test was 0.484 (Figure 3(Ba)). The unpaired Cohen’s d between the control and the loading (+)-48 h incubation group was 0.631 (95.0% CI −0.313, 1.68), and the *p*-value of the two-sided permutation *t*-test was 0.133 (Figure 3(Bb)) (each subgroup/time point, *n* = 12).

Compressive loading for 24 and 48 h tended to increase the production of both ALP and osteocalcin by osteoblasts in comparison with those from non-loaded osteoblasts (Figure 3A,B), suggesting that continuous and physiologic mechanical loading induced osteoblast differentiation.

In contrast with osteoblasts, no significant differences in the production of the articular cartilage components, type II collagen and proteoglycan, by chondrocytes were observed between the mechanically stressed chondrocyte group and the non-stressed chondrocyte group (Figure 4A,B).

Regarding the level of proteoglycan production from chondrocytes, the unpaired Cohen’s d between the control and the loading (+)-24 h incubation group is −0.323 (95.0% CI −1.31, 0.547) and the *p*-value of the two-sided permutation *t*-test was 0.425 (Figure 4(Aa)). The unpaired Cohen ‘s d between the control and the loading (+)-48 h incubation group was −0.788 (95.0% CI −1.65, 0.159), and the *p*-value of the two-sided permutation *t*-test was 0.0676 (Figure 4(A-b)) (each subgroup/time point, *n* = 12).

In the level of type II collagen production from chondrocytes, the unpaired Cohen’s d between the control and the loading (+)-24 h incubation group was 0.0315 (95.0% CI −0.887, 0.87) and the *p*-value of the two-sided permutation *t*-test was 0.942 (Figure 4(Ba)). The unpaired Cohen’s d between the control and the loading (+)-48 h incubation group was −0.37 (95.0% CI −1.47, 0.712), and the *p*-value of the two-sided permutation *t*-test was 0.41 (Figure 4(Bb)) (each subgroup/time point, *n* = 12). These findings suggested that physiologic mechanical stress may not influence chondrocyte activity in articular cartilage tissue.

### 2.3. Effects of Mechanical Loading on Glut1, SIRT1, and Runx2 Expressions in Osteoblasts and Chondrocytes

To determine whether physiological mechanical loading regulates the expression of regulatory factors governing the energy metabolism of a transcription factor for osteoblast differentiation, we examined the expressions of the glucose transporter Glut1, the cellular energy sensor SIRT1, and the osteogenic transcription factor Runx2 in the presence or absence of continuous physiologic loading in osteoblasts and chondrocytes. Runx2 is also well known as an inducer (transcription factor) of chondrocyte maturation as well as an osteogenic transcription factor for osteoblast differentiation [30].

The expression of Glut1 in compressive mechanically loaded osteoblasts was significantly higher than that in non-loaded osteoblasts after 48 h incubation of the 3D osteoblast–collagen sponge construct (Figure 5A,B, control osteoblast vs. mechanically loaded osteoblast: *p* = 0.0019).

In contrast with the upregulated expression of Glut1, compressive loading significantly decreased the expression of SIRT1 in osteoblasts in comparison with non-loaded osteoblasts after 48 h incubation (Figure 5C,D, control osteoblast vs. mechanically loaded osteoblast: *p* = 0.0133). The expression of Runx2 in the compressive-loaded osteoblasts tended to be higher than in non-loaded osteoblasts (Figure 5E,F).

In contrast with osteoblasts, no significant differences in the expressions of Glut1, SIRT1, or Runx2 were observed between the mechanically loaded chondrocytes and the non-loaded chondrocytes after 24 h of incubation in the 3D chondrocyte–collagen sponge construct (Figure 6). Although there were significant differences for chondrocytes tendencies for increased SIRT1 and Runx2 with mechanical loading, no significant differences were observed between the mechanically loaded chondrocytes group and the non-loaded chondrocytes group (*p* > 0.05, *n* = 12 collagen sponge samples per subgroup × three independent experiments).

### 2.4. Effects of Glut1 Inhibitor on Mechanical Loading-Induced Bone Formation In Vitro

To determine whether Glut1 is involved in the response to mechanical loading, we compared protein levels of SIRT1 and Runx2 in the control (no loading) and mechanically loaded osteoblasts in the presence or absence of WZB117.

The expression of Glut1 in osteoblasts was significantly increased by the mechanical loading in medium only group (Figure 7(Aa,Ad), * *p* = 0.0162) and the control group (DMSO + medium, Figure 7(Ab,Ad), * *p* = 0.0066). The WZB117-treated group showed a lower level of expression of Glut1 than the control group. Mechanical loading did not induce the expression of Glut1 in the WZB117-treated osteoblasts (Figure 7(Ac,Ad), *p* > 0.05).

Interestingly, although the expression of SIRT1 was significantly inhibited by mechanical loading in osteoblasts of the medium-only group (Figure 7(Ba,Bd), * *p* = 0.0028) and the control group (Figure 7(Bb,Bd), * *p* = 0.0372), no significant difference in the expression of SIRT1 in osteoblasts was observed even in the presence of mechanical loading in the WZB117-treated group (Figure 7(Bc,Bd), *p* > 0.05).

Regarding the expression of Runx2 after 48 h incubation, compressive mechanical loading significantly increased the expression of Runx2 in the osteoblasts of the medium-only group (Figure 7(Ca,Cd), * *p* = 0.0098) and the control group (Figure 7(Cb,Cd), * *p* < 0.0001), in comparison with non-loaded osteoblasts. In contrast, the expression of Runx2 in osteoblasts remained unchanged by mechanical loading in the WZB117-treated group (Figure 7(Cc,Cd), *p* > 0.05).

## 3. Discussion

A greater understanding of the cellular responses to mechanical stress in bone and cartilage metabolism is conducive to gaining new insights into the pathophysiology and development of novel therapies for age-related bone and joint diseases, including osteoporosis and OA.

Osteocytes, entrapped within a mineralized bone matrix, play a central role in directing the bone response to physiologic mechanical loading, leading correspondingly to bone formation [13,14]. Recent studies clearly indicated an important role of the osteocyte-produced protein sclerostin as a key mediator of the molecular mechanism involved in mechanical stress-induced bone formation [15,16,48,49]. It has been demonstrated that mechanical stress inhibits the activity of sclerostin, leading to activation of Wnt/beta-catenin signaling in osteocytes and osteoblasts, resulting in osteoblast differentiation and bone formation [14,15,16,49]. Cellular interaction between osteocytes and osteoblasts, via the sclerostin-Wnt/beta-catenin pathway, is required for mechanical loading-induced osteoblast differentiation following acceleration of bone formation (Figure 8A).

In the present study, our results clearly indicate that physiologic and compressive mechanical loading induces the upregulation of Glut1 and the downregulation of SIRT1 in osteoblasts but not in chondrocytes. These findings suggest that mechanical loading increases glucose uptake via an upregulated Glut1 in osteoblasts. This increase in glucose uptake may reduce the activity of the energy sensor SIRT1 as well as AMPK, since SIRT1 is known to crosstalk with AMPK, forming a SIRT1–AMPK positive feedback loop in cellular energy metabolism [23,50,51]. From the results of previous studies, we postulated that the mechanical loading could induce the increased level of Glut-1 and its resultant decreases in both energy sensors, SIRT1 and AMPK. As shown in Figure 8, the main purpose of this study was to clarify the mechanism of the loading-induced bone formation (osteogenesis) with a sclerostin-Wnt/beta-catenin-independent signaling pathway. Since it is known that SIRT1 but not AMPK negatively regulates the level of Runx2 (a master transcription factor for chondrocyte hypertrophy, osteogenic, and chondrogenic differentiation), we focused on SIRT1 among SIRT1 and AMPK. We conclude that mechanical stress suppresses the level of SIRT1 in response to the stress-induced upregulation of Glut1 in osteoblasts (Figure 8B). This is the first report of an interaction between mechanical stress and cellular energy metabolism, especially from the point of view of the energy sensor SIRT1.

The transcription factor Runx2 is a master transcription factor for osteogenic and chondrogenic differentiation [30,31]. Its expression precedes osteogenic and chondrogenic differentiation of mesenchymal stem cells [30,31,52]. The interaction of Runx2 with mechanical stress, however, has remained poorly understood. Recently, it has been demonstrated that Runx2 activity is negatively regulated by SIRT1 in human somatic cells, such as vascular smooth muscle cells and chondrocytes [53,54,55]. Takemura et al. demonstrated that SIRT1 has an inhibitory effect on Runx2 expression and that the osteoblastic change in vascular smooth muscle cells is at least in part dependent on SIRT1 through a mechanism involving the SIRT1–Runx2 pathway [53]. Jeon et al. have also shown that the acetylation of Runx2 is important in osteoblast differentiation and is downregulated by SIRT1 [55]. They concluded that SIRT1 can deacetylate Runx2, resulting in the inhibition of Runx2-related osteoblastic transition in vascular smooth muscle cells. In addition, it has been demonstrated that the overexpression of SIRT1 inhibits Runx2 gene expression in human chondrocytes [56]. These findings suggest that SIRT1 has the potential to suppress Runx2 activity in cells.

Our results in the present study show that mechanical stress induced upregulation of the glucose transporter Glut1, downregulation of the energy sensor SIRT1, and upregulation of the osteogenic factor Runx2 in osteoblasts but not in chondrocytes. Mechanical stress-induced downregulation of SIRT1 may lead to the upregulation of Runx2, resulting in osteoblast differentiation. Our data of mechanical loading-suppressed SIRT1 and -activated Runx2 in osteoblasts are consistent with previous findings regarding interactions between SIRT1 and Runx2 [31,53,54,55,56]. Furthermore, incubation with a Glut1 inhibitor blocked the mechanical loading-induced changes in SIRT1 and Runx2, suggesting that Glut1 is necessary to mediate the responses of SIRT1 and Runx2 to mechanical loading. We conclude that mechanical stress suppresses the activity of the energy sensor SIRT1 via stress-induced activation of Glut1, resulting in the upregulation of Runx2 in osteoblasts. Indeed, our data clearly demonstrate that mechanical loading increased the production of osteocalcin and ALP in osteoblasts. This means that mechanical loading-induced Runx2 activation, in response to SIRT1 suppression via Glut1 activation, accelerates the production of osteocalcin and ALP, indicating the induction of osteoblast differentiation and bone formation (Figure 8B). As shown in Figure 2B,D, continuous and physiologic mechanical loading induced the deposition of microgranules on the surface of collagen sponge fibers as well as on the cell surface of osteoblasts. The levels of microgranules deposition increased with incubation time on both surface of osteoblasts and collagen fibers, suggesting the production of calcium deposits and calcium mineralization by osteoblasts. However, in the present study, the elemental composition of microgranules still remains unknown. To confirm the composition of microgranules, we plan to analyze the composition of microgranules and calcium mineralization by alizarin red staining. We have also considered the difference in the mechanical stress response among the continuous/physiologic loading, the cyclic loading, and the long-term loading. Chen XI et al. demonstrated that cyclic compression load promoted osteoblast differentiation and maturation, leading to bone formation [49]. We also plan to study the osteoblast activity and the calcium mineralization in response to the cyclic stress using a newly developed cyclic loading 3D-cell culture system.

In contrast with osteoblasts, the level of mechanical stress that induces a response in osteoblasts appears not to induce a reaction in chondrocytes. Articular cartilage is an avascular tissue that has a poor spontaneous self-healing capacity. Adult chondrocytes into the cartilage tissue have little or no potential of proliferation. In contrast, bone tissue with a rich vascular system has the potential to remodel with a high level of regeneration. To maintain the function of articular cartilage in the joint, chondrocytes might have a mechanical stress tolerance to some degree. Although further studies are needed to verify the difference in the mechanical stress response between osteoblasts and chondrocytes, there might be some different mechanism in the stress tolerance between the two.

In conclusion, our present data indicate that the activation of Glut1 is required for mechanical stress-induced bone formation via the signal transduction network between the cellular energy sensor SIRT1 and the osteogenic transcription factor Runx2 in osteoblasts. Our findings indicate that there is a sclerostin–Wnt/beta-catenin-independent signaling mechanism connecting osteocytes and osteoblasts for osteoblast differentiation and bone formation (Figure 8A,B). Mechanical loading may directly induce osteoblast differentiation and bone formation without a sclerostin–Wnt/beta-catenin-dependent signaling mechanism connecting osteocytes and osteoblasts.

## 4. Materials and Methods

### 4.1. Monolayer Human Cell Cultures: Chondrocytes and Osteoblasts

Osteoblasts: Human osteoblasts were purchased from PromoCell GmbH (Heidelberg, Germany) and cultured according to the recommendations of the supplier in Dulbecco’s modified Eagle’s medium (DMEM, Sigma-Aldrich Japan KK., Tokyo, Japan) containing 10% fetal bovine serum (FBS: Fujifilm Wako Pure Chemical Inc., Tokyo, Japan), 2 mM L-glutamine (Fujifilm Wako Pure Chemical Inc.), and 100 U/mL each of penicillin and streptomycin (Penicillin/Streptmycin solution, Thermo Fisher Scientific Inc., Waltham, MA, USA) at 3 °C in a humidified atmosphere of 95% air and 5% CO_2_.

Chondrocytes: Human articular cartilage tissues were obtained from the knee joint of an OA patient who underwent arthroplastic knee surgery and who provided written informed consent before participating in the study (female, 67 years old). The study protocol was reviewed and approved by the local Ethics Committee of St. Marianna University School of Medicine (permission number: 1315). The procedures followed were in accordance with the ethical standards of the Ethics Committee and with the Helsinki Declaration of 1975, as revised in 2000.

Articular cartilage explants were cut into small pieces, washed with PBS, and digested individually with 1.0 mg/mL collagenase type I (#035-17604, Fujifilm Wako Pure Chemical Inc., Tokyo, Japan) in DMEM (Sigma-Aldrich) overnight on a shaking platform at 3 °C. Isolated chondrocytes from each explant were collected following centrifugation; washed three times with PBS; resuspended; and then cultured in DMEM supplemented with 10% heat-inactivated FBS, 2 mM L-glutamine (Sigma-Aldrich), and 100 U/mL each of penicillin and streptomycin (Cosmo Bio) at 3 °C in a humidified atmosphere of 95% air and 5% CO_2_, as previously reported [23,30,57]. A sufficient number of cultured chondrocytes were obtained from the relatively normal parts of surgically resected cartilage tissues.

### 4.2. Generation of a 3D Cell–Collagen Scaffold Construct for Cultured Human Cells

Three-dimensionally cultured tissue was generated from human chondrocytes or osteoblasts according to previously described methods [58,59]. Cultured human cells in monolayer were collected and resuspended at 2.0 × 10^6^ cells/mL in DMEM. The cell suspension was seeded onto a collagen sponge scaffold in a 96-well cell culture plate (AGC Techno Glass Co., Ltd., Shizuoka, Japan) at a density of 1 × 10^5^ cells/scaffold/well and incubated at 37 °C in a humidified atmosphere of 95% air and 5% CO_2_ to produce the 3D cell–collagen scaffold construct. A porous collagen sponge (5 mm diameter, 3 mm thick) was purchased from AteloCell^®^ (Koken Co., Ltd., Tokyo, Japan). The pore sizes were designed to be 200 to 400 µm. After gelation of the 3D cell–collagen scaffold construct, the 3D culture tissues were incubated overnight in growth medium for osteoblasts or chondrocytes, as appropriate, at 37 °C in a 5% CO_2_ atmosphere.

### 4.3. Mechanical Loading of the 3D Cell Culture Construct

The 3D cell-cultured constructs described above were placed into individual wells of a 12-well culture dish (Iwaki) and maintained in 1.5 mL DMEM. Then, mechanical loading was applied to the 3D culture constructs with the use of a custom-designed and -built apparatus, as shown in Figure 1A,B. Continuous physiologic compression was applied to the tested 3D constructs at 0 kPa (non-loaded) or 25.5 gf/cm^2^ (2.5 kPa) in reference to previous studies [58,59]. The level of mechanical loading studied in the present study did not induce the destruction of the collagen sponge. After removing the weight (mechanical loading), the compressed collagen sponges were restored. Mechanical loading experiments were carried out for 24 and/or 48 h in a humidified incubator maintained at 37 °C in 5% CO_2_ atmosphere.

### 4.4. Scanning Electron Microscopy

Scanning electron microscopy (SEM) analysis was performed to examine the surface structure of the 3D culture constructs. After serial fixations of scaffold samples with 2% glutaraldehyde, 1% osmium tetroxide, and 1% tannic acid aqueous solution, the surface morphology of the scaffolds was observed by SEM (S-4800, Hitachi Ltd., Tokyo, Japan). 

In SEM images, the number of microgranules per 10 cells/one collagen sponge sample (*n* = 12 samples/subgroup/each time point in a 12-well culture dish) was counted by three independent researchers. The analysis was triplicated, and then, the mean number of microgranules per one cell surface was calculated.

### 4.5. Cellular Protein Content in the 3D Cell-Culture Construct

The total protein content of the cells in each 3D cell–collagen sponge construct was extracted by the following method. The 3D culture constructs were cut into small pieces, washed with PBS and digested with 1.5 mg/mL collagenase B (Sigma-Aldrich) in DMEM at 3 °C overnight on a shaking platform. The isolated cells were centrifuged, washed three times with PBS, and collected for in vitro analyses. Protein samples were collected from the cells for each in vitro assay, as previously described [31,57].

### 4.6. Effects of Mechanical Loading on Cellular Activities In Vitro

In osteoblasts and chondrocytes, to verify the involvement of mechanical loading in each cellular activity and in the levels of expressions of Glut1, SIRT1, and Runx2 in each cell strain, we analyzed the level of each cell strain-specific factor after mechanical loading of the 3D cell–collagen sponge construct. After loading of the 3D cell–collagen sponge construct, the cellular protein and conditioned medium were collected from the 3D culture construct.

(i) Osteoblast activity: The levels of ALP and osteocalcin produced by osteoblasts were measured using ELISA kits (ALP assay kit; Bio Vision Inc., Milpitas, Japan; osteocalcin assay kit; Takara Bio Inc., Kusatsu, Japan).

(ii) Chondrocyte activity: The levels of proteoglycan and type II collagen produced by chondrocytes were measured using ELISA kits (Proteoglycan assay kit; Thermo Fisher Scientific, Tokyo, Japan; type II collagen assay kit; R&D Systems Inc., Minneapolis, MN, USA).

To study the effects of mechanical stress on the expressions of Glut1, SIRT1, and Runx2 in each cell strain of osteoblasts and chondrocytes, the expression levels of Glut1, SIRT1, and Runx2 in cells were analyzed by Western blotting as previously described [23,31,57].

(iii) Expressions of Glut1, SIRT1, and Runx2 in each cell strain: For Western blotting, the antibodies used were a polyclonal antibody against human Glut1 (1:2000 dilution; Abcam Inc., Cambridge, UK), polyclonal antibody against human SIRT1 (1:5000 dilution; Abcam Inc.), Runx2 (#FA2006, 1:1000 dilution; R&D Systems Inc.), β-actin (1:5000 dilution; Fujifilm Wako), and the corresponding secondary antibody conjugated with horseradish peroxidase (Aglient Technologies Japan, Tokyo, Japan, rabbit IgG p0448 for anti-Glut1 antibody (1:10,000 Dilution) and anti-SIRT1 antibody (1:5000 dilution), or goat IgG p0449 for anti-Runx2 antibody (1:20,000 dilution)). The antibody-bound protein bands were visualized, and densitometry of the signal bands was analyzed using an extended cavity laser system (GE Healthcare Bio-sciences KK, Tokyo, Japan). The data are representative of three–four independent experiments (each subgroup/time point, *n* = 12).

### 4.7. Effects of Glut1 Inhibitor on Mechanical Loading-Induced Bone Formation In Vitro

To clarify whether Glut1 is involved in each cell response to mechanical loading, we compared the osteoblastic activity and protein levels of SIRT1 and Runx2 in the control (no loading) and mechanically loaded osteoblasts in the presence or absence of a Glut1 inhibitor (WZB117; Selleck Biotech Ltd. Inc., Tokyo, Japan). Previous reports demonstrated that approximately 10 μM of WZB117 seems to be suitable for the Glut-1 inhibition assay with a Glut-1 inhibitor. We also pre-studied the suitable concentration of WZB117 in the pilot study and chose 10 μM of WZB117 for the Glut-1 inhibition analysis.

As described above, to produce the 3D cell-cultured constructs, human chondrocytes or osteoblasts were seeded onto a porous collagen sponge scaffold in a 96-well cell culture plate and incubated overnight, individually. On the next day, one round 3D collagen scaffold disc was placed into each well of a 12-well culture dish and maintained in 1.5 mL culture medium in the presence or absence of 10.0 µM of the Glut1 inhibitor (WZB117). Solutions of WZB117 were freshly prepared by dissolving the compound in dimethyl sulfoxide (DMSO) before each experiment. The culture medium-only group and the control group (DMSO solution + culture medium) were prepared as controls for the Glut1 inhibitor-treated group (WZB117 in DMSO solution + culture medium). Then, in the presence or absence of WZB117, mechanical loading was applied to the 3D culture constructs for 48 h in the same manner as described above. The data are representative of three–four independent experiments (each subgroup/time point, *n* = 12). After loading, the expressions of Glut 1, SIRT1, and Runx2 of each cell strain were evaluated as described above.

### 4.8. Statistical Analysis

The results of each experimental condition were determined from the mean of triplicate experiments. The data are expressed as means ± standard deviation. For parametric data sets, a one-way analysis of variance (ANOVA) with Bonferroni multiple comparison post hoc test was used. The significance level was set at *p* < 0.05.

## Figures and Tables

**Figure 1 ijms-22-09070-f001:**
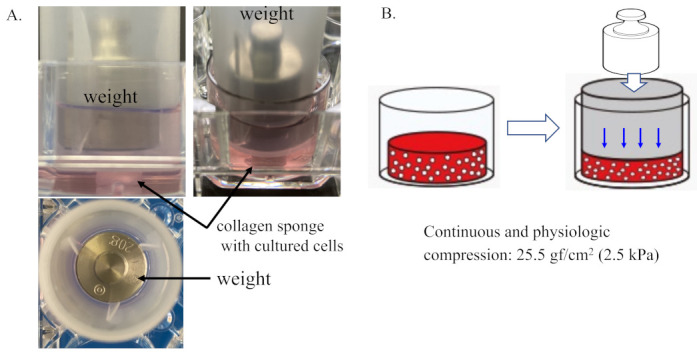
Three-dimensional cell–collagen scaffold construct. Continuous and physiologic compression was applied to the tested 3D constructs at 0 kPa (non-loaded) or 25.5 gf/cm^2^ (2.5 kPa). (**A**) Representative images of the generation of the 3D cell–collagen scaffold construct for human cultured cells. (**B**) Scheme showing mechanical loading of the 3D cell culture construct.

**Figure 2 ijms-22-09070-f002:**
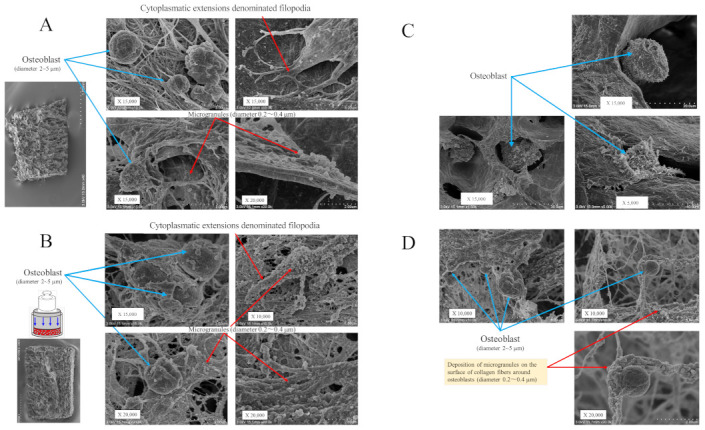
Scanning electron microscopy (SEM) of the 3D cell–collagen scaffold construct. Representative SEM images: (**A**) the 3D osteoblast–collagen scaffold construct (non-loading), (**B**) the 3D osteoblast–collagen scaffold construct (after 24 h mechanical loading), (**C**) the 3D osteoblast–collagen scaffold construct (after 48 h non-loading), and (**D**) the 3D osteoblast–collagen scaffold construct (after 48 h mechanical loading). (**E**) Representative SEM images of a 3D chondrocyte–collagen scaffold construct (non-loading), (**F**) the 3D chondrocyte–collagen scaffold construct (after 24 h mechanical loading), (**G**) the 3D chondrocyte–collagen scaffold construct (48 h non-loading), and (**H**) the 3D chondrocyte–collagen scaffold construct (after 48 h mechanical loading) (each subgroup/time point, *n* = 12).

**Figure 3 ijms-22-09070-f003:**
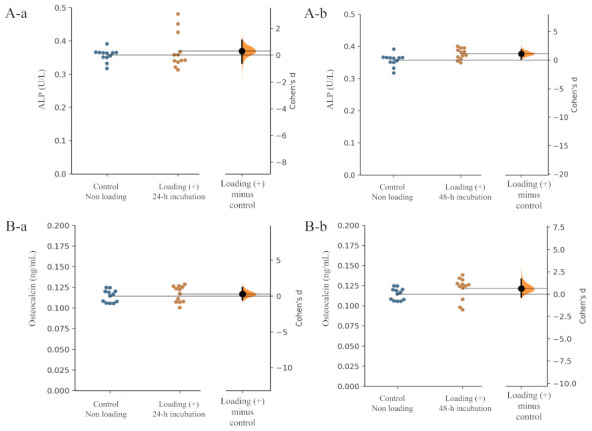
Effects of mechanical loading on osteoblast activity. Cohen’s d between the control and the mechanical loading group is shown in the above Gardner–Altman estimation plot. Both groups are plotted on the left axes; the mean difference is plotted on a floating axes on the right as a bootstrap sampling distribution. The mean difference is depicted as a dot; the 95% confidence interval is indicated by the ends of the vertical error bar. The effect sizes and CIs are reported above as effect size (CI width lower bound; upper bound). (**A**) ALP: (**a**) control group versus mechanical loading (+)-24 h incubation group, (**b**) control group versus mechanical loading (+)-48 h incubation group. (**B**) Osteocalcin: (**a**) control group versus mechanical loading (+)-24 h incubation group, (**b**) control group versus mechanical loading (+)-48 h incubation group).

**Figure 4 ijms-22-09070-f004:**
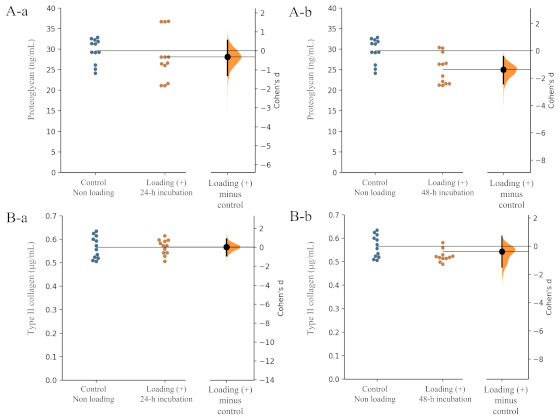
Effects of mechanical loading on chondrocyte activity. Effects of mechanical loading on osteoblast activity. Cohen’s d between the control and mechanical loading group is shown in the above Gardner–Altman estimation plot. Both groups are plotted on the left axes; the mean difference is plotted on a floating axes on the right as a bootstrap sampling distribution. The mean difference is depicted as a dot; the 95% confidence interval is indicated by the ends of the vertical error bar. The effect sizes and CIs are reported above as effect size (CI width lower bound; upper bound). (**A**) Proteoglycan: (**a**) control group versus mechanical loading (+)-24 h incubation group, (**b**) control group versus mechanical loading (+)-48 h incubation group. (**B**) Type II collagen: (**a**) control group versus mechanical loading (+)-24 h incubation group, (**b**) control group versus mechanical loading (+)-48 h incubation group).

**Figure 5 ijms-22-09070-f005:**
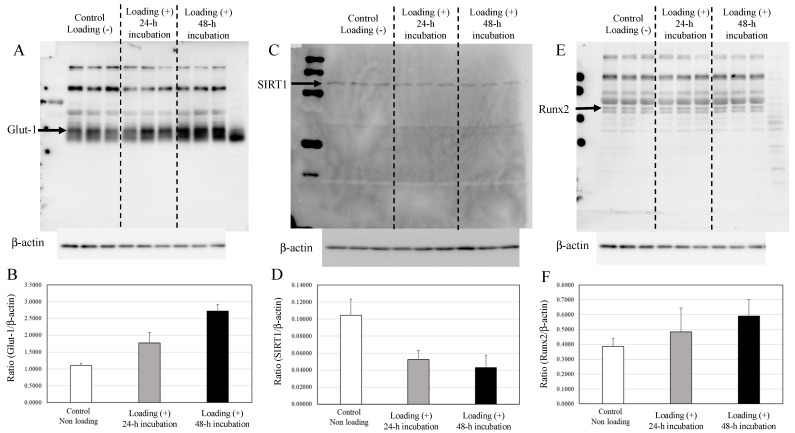
Expressions of Glut1, SIRT1, and Runx2 in osteoblasts. (**A**) The expression of Glut1 in non-loaded osteoblasts, 24 h loaded osteoblasts, and 48 h loaded osteoblasts. (**B**) The expression relative to β-actin. (**C**) The expression of SIRT1 in non-loaded osteoblasts, 24 h loaded osteoblasts, and 48 h loaded osteoblasts. (**D**) The expression relative to β-actin. (**E**) The expression of Runx2 in non-loaded osteoblasts, 24 h loaded osteoblasts, and 48 h loaded osteoblasts. (**F**) The expression relative to the level of β-actin.

**Figure 6 ijms-22-09070-f006:**
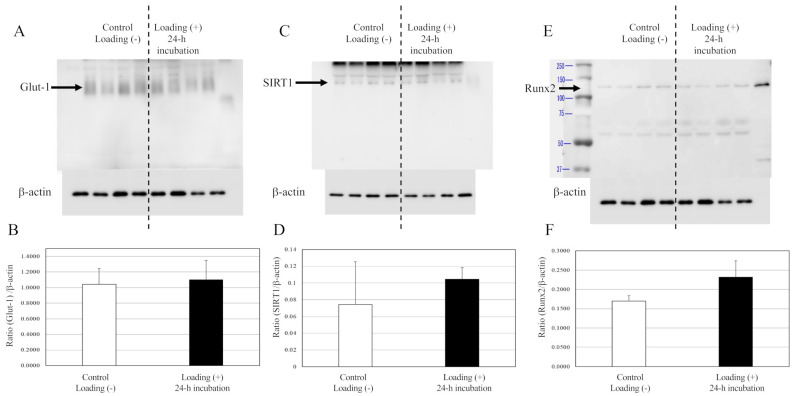
Expressions of Glut1, SIRT1, and Runx2 in chondrocytes. The expression of Glut1s (**A**), SIRT1 (**C**), and Runx2 (**E**) in chondrocytes were not affected by loading (Western blot). (**B**,**D**,**F**) The expressions relative to the levels of β-actin. (each subgroup/time point, *n* = 12 × three independent experiments).

**Figure 7 ijms-22-09070-f007:**
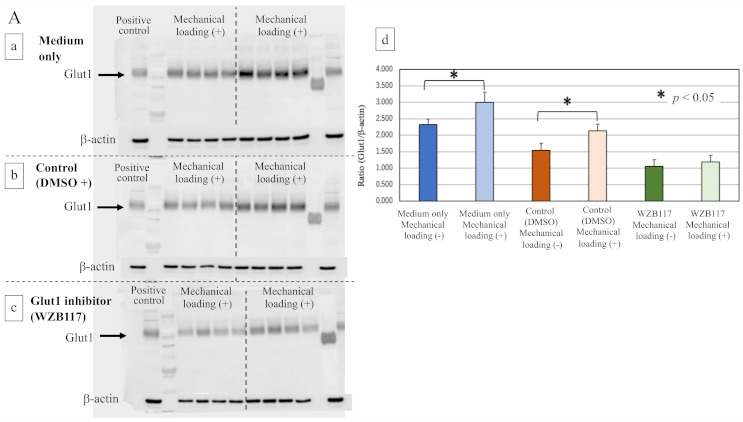
Effects of the Glut1 inhibitor on Glut-1, SIRT1, and Runx2 in osteoblasts. (**A**) The expression of Glut1 in mechanically loaded or non-loaded osteoblasts in the presence or absence of a Glut1 inhibitor (Western blot). (**a**) The culture medium only, (**b**) the control group (DMSO solution + culture medium), (**c**) the Glut1 inhibitor (WZB117)-treated group (10.0 μM WZB117 in DMSO solution + culture medium), and (**d**) the expression relative to β-actin. (**B**) The expression of SIRT1 in mechanically loaded or non-loaded osteoblasts in the presence or absence of a Glut1 inhibitor (Western blot). (**a**) The culture medium only, (**b**) the control group (DMSO solution + culture medium), (**c**) the Glut1 inhibitor (WZB117)-treated group (10.0 μM WZB117 in DMSO solution + culture medium), and (**d**) the expression relative to β-actin. (**C**) The expression of Runx2 in mechanically loaded or non-loaded osteoblasts in the presence or absence of a Glut1 inhibitor (Western blot). (**a**) The culture medium only, (**b**) the control group (DMSO solution + culture medium), (**c**) the Glut1 inhibitor (WZB117)-treated group (10.0 μM WZB117 in DMSO solution + culture medium), and (**d**) the expression relative to β-actin. * *p* < 0.05.

**Figure 8 ijms-22-09070-f008:**
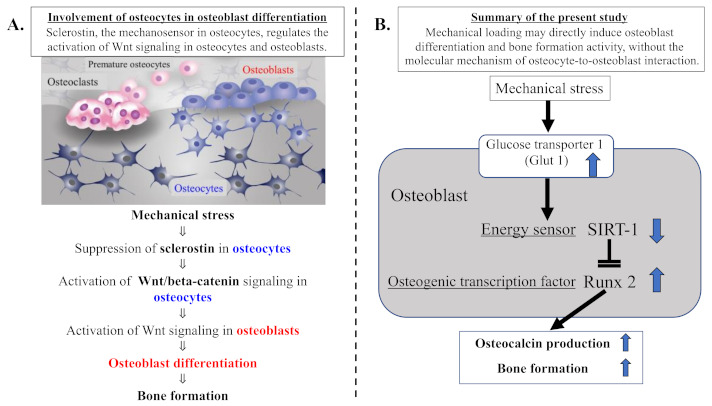
Summary of our current study. (**A**) Involvement of osteocytes in osteoblast differentiation. Mechanical loading suppresses the expression of osteocytic sclerostin following activation of Wnt/beta-catenin signaling in osteocytes and osteoblasts, resulting in osteoblast differentiation and bone formation activity. Cellular interaction of osteoblasts with osteocytes, via the sclerostin–Wnt/beta-catenin signaling pathways, plays an important part in physiologic mechanical stress-mediated bone metabolism. (**B**) Mechanical loading may directly induce osteoblast differentiation and bone formation activity, without the molecular mechanism of osteocyte-to-osteoblast interaction. The mechanical stress-induced expression of Glut1 and resultant uptake of glucose may suppress the level of the energy sensor SIRT1 in osteoblast energy metabolism. Since SIRT1 is recognized to negatively regulate Runx2 activity, the mechanical stress-induced suppression of SIRT1 results in increased activity of the osteogenic transduction factor Runx2 in osteoblasts, leading to osteoblast differentiation and bone formation.

## Data Availability

The data that support the findings of this study are available from the corresponding author, (Kazuo Yudoh), upon reasonable request.

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
