# Peer review of "Physiologic Mechanical Stress Directly Induces Bone Formation by Activating Glucose Transporter 1 (Glut 1) in Osteoblasts, Inducing Signaling via NAD+-Dependent Deacetylase (Sirtuin 1) and Runt-Related Transcription Factor 2 (Runx2)"

_ijms, 2021, doi:10.3390/ijms22169070_

Round 1

Reviewer 1 Report

The paper deals with the mechanisms of osteogenic differentiation induction by mechanical stress and the role of the Glut-1 – Sirt-1 – Runx 2 axis.

General

  • The title is very long

Abstract

  • Abstract too long, some redundant parts
  • Very long sentences at the beginning

Introduction

  • What is known about the general role of Sirt-1 in osteoblasts?
  • More information on the three proteins in chondrocytes, why also chosen as cellular model?
  • Some passages are repetitive, this could be shortened (e.g. line 113-116)

Material and Methods

  • Please indicate in the figure legends or somewhere else where parametric and where non-parametric statistical tests were used
  • Where the used chondrocytes all from the same OA patient? Could the “OA character” of the cells influence the results?

Results

  • The 3 of 3D in the title of the first subsection has moved to the subsection numbering (“2.1.3. D cell−“)
  • Line 134: test should be tested
  • Figure 2: How was the quantification of microgranules on SEM Images done?
  • Was the scaffold structure affected by the mechanical loading?
  • Please add an SEM-image of the 48h non-loaded control (can also be in the supplementary).
  • Figure 3: There is an error in the y-axis labeling of osteocalcin (“osteocalxin”).
  • Figure 4: What happens when chondrocytes where incubated with mechanical loading for 48h?
  • Figure 5: Please explain why GAPDH was chosen as housekeeping gene. Normalization of glucose dependent proteins on GAPDH is normally considered difficult.
  • Figure 6 = twice
  • Figure 6: It seems like there is also for chondrocytes a tendency for increased Runx2 with loading. Please mention this in the text.
  • 5: Please add the role of WZB117 (Glut-1 inhibitor)
  • Figure 7: One of the two controls could get into the supplementary, two controls are rather confusing.
  • Figure 7: Why was here normalized to β-Actin levels? Is β-Actin a good housekeeper when working with mechanical loading.
  • Does incubation with Glut-1 inhibitor also influence AP and osteocalcin levels?
  • Figure 5+6+7: Please explain in the figure legend which exemplary bands of the blots are shown and what the frames mean. Additionally, please provide the full western blot images.
  • What about AMPK levels? It is proposed that the connection between Glut-1 and Sirt-1 can be mediated by AMPK.
  • The β in β-Actin is replaced by some error sign in the manuscript (e.g. figure 7 figure legend).

Discussion

  • Some repetitive passages (line 325-327; 341-345)
  • What happens with cyclic mechanical stress or longer incubation with the weight?
  • Please discuss more why mechanical loading does not induce chrondrogenic differentiation. This is contrary to the common sense.

Author Response

Answer letter to Reviewer 1

 General

  • The title is very long

Abstract

  • Abstract too long, some redundant parts
  • Very long sentences at the beginning

Answer to three comments

Thank you very much for your helpful comment. We agree with your comments.

         As you comment, we have carefully checked and have shortened the “title”, “abstract” section and all sections without loss of our point. We have revised the title and abstract section in the revised manuscript (please find them in the revised manuscript). Thank you again for your comment.

Introduction

  • What is known about the general role of Sirt-1 in osteoblasts?

Answer:

We appreciate your comment. Previous studies demonstrated that Sirt-1 has two important roles in “regulations of the cellular energy metabolism and the cellular activity/differentiation” and “stress tolerance” in a variety of cells including osteoblasts (references 21,24,25). Regarding osteoblasts, in addition to the roles in “cellular energy metabolism” and “stress response (tolerance)”, it has been suggested that Sirt-1 promotes osteogenesis, osteoblast differentiation and proliferation through two independent key regulators, "Runx2" and "FoxOs" as the following:

  (Sirt-1-Runx2 interaction in osteoblasts)

Summary:  It has been reported that Runx2, osteogenic transcription factor, is required for chondrocyte hypertrophy and osteogenesis (osteoblast differentiation/proliferation) in the bone and joint tissue (i, ii,iii). Several studies demonstrated that Sirt-1 activator mediates the expression of Runx2 in mesenchymal stem cells (iv, v) and that Sirt-1 inactivation decreased the expression of Runx2 in chondrocytes (vi).

  1. Wang X, Manner PA, Horner A, Shum L, Tuan RS, Nuckolls GH. Regulation of MMP-13 expression by RUNX2 and FGF2 in osteoarthritic cartilage. Osteoarthritis Cartilage 12(12), 963-973, 2004.
  2. Kawaguchi H. Endochondral ossification signals in cartilage degradation during osteoarthritis progression in experimental mouse models. Mol Cells 25(1), 1-6, 2008.
  • Kamekura S, Kawasaki Y, Hoshi, K, Shimoaka T, Chikuda H, Maruyama Z, Komori T, Sato S, Takeda S, Karsenty G, Nakamura K, Chung UI, Kawaguchi H. Contribution of runt-related transcription factor 2 to the pathogenesis of osteoarthritis in mice after induction of knee joint instability. Arthritis Rheum 54(8), 2462-2470, 2012.
  1. Kim HJ, Braun HJ, Dragoo JL. The effect of resveratrol on normal and osteoarthritic chondrocyte metabolism. Bone Joint Res 3(3), 51-59, 2014.
  2. Shakibaei M, Shayan, P,Busch, F, Aldinger, C, Buhrmann C, Lueders, C, Mobasheri A. Resveratrol mediated modulation of Sirt-1/Runx2 promotes osteogenic differentiation of mesenchymal stem cells: potential role of Runx2 deacetylation. PLOS One 7(4), e35712, 2012.
  3. Terauchi K, Kobayashi H, Yatabe K, Yui N, Fujiya H, Niki H, Musha H, Yudoh K. The NAD-Dependent Deacetylase Sirtuin-1 Regulates the Expression of Osteogenic Transcriptional Activator Runt-Related Transcription Factor 2 (Runx2) and Production of Matrix Metalloproteinase (MMP)-13 in Chondrocytes in Osteoarthritis. Int J Mol Sci 7(7), pii: E1019. doi: 10.3390/ijms17071019, 2016.

These findings indicate that Sirt-1 promotes the osteogenesis through the mechanism involving the Sirt-1 - Runx2 interaction.

  (Sirt-1-FoxOs interaction in osteoblasts) 

Summary:  Transcription factors, FoxO1, FoxO3, FoxO4 and FoxO6, represent a subclass of a large family of forkhead proteins which are characterized by the presence of a winged-helix DNA binding domain, Forkhead box (i). It has been demonstrated that Sirt-1 deacetylates a transcription factor, FoxOs (ii).

         More recently, FoxOs (1~4) are known to control bone resorption and formation (iii-viii). Iyer S. et al. reported that acetylation of FoxOs controls the interaction between FoxOs and β-catenin, while Sirt-1-mediated FoxOs deacetylation prevents this interaction and potentiates Wnt signaling, resulting in the induction of the osteoblast proliferation (vii, viii). They concluded that FoxOs may have a role as an osteoblast progenitor (transcription factor) and Sirt1-FoxOs interaction participates in the osteoblast differentiation.

  1. Eijkelenboom A, Burgering BM, FOXOs: signalling integrators for homeostasis maintenance, Nat Rev Mol Cell Biol. 14(2):83–97, 2013.
  2. Wagner GR, Hirschey MD, Nonenzymatic protein acylation as a carbon stress regulated by sirtuin deacylases, Mol Cell 54(1): 5–16, 2014.

iii. Iyer S, Ambrogini E, Bartell SM, Han L, Roberson PK, Cabo R, Jilka RL, Weinstein RS, O'Brien CA, Manolagas SC, Almeida M, FoxOs attenuate bone formation by suppressing Wnt signaling, J Clin Invest 123(8) (2013) 3404–3419.

  1. Bartell SM, Kim HN, Ambrogini E, Han L, Iyer S, Serra US, Rabinovitch P, Jilka RL, Weinstein RS, Zhao H, O'Brien CA, Manolagas SC, Almeida M, FoxO proteins restrain osteoclastogenesis and bone resorption by attenuating H2O2 accumulation, Nat Commun 5 (2014) 3773.
  2. Tan P, Guan H, Xie L, Mi B, Fang Z, Li J, Li F, FOXO1 inhibits osteoclastogenesis partially by antagnozing MYC, Sci Rep 5 (2015) 16835.
  3. Kim HN, Han L, Iyer S, de Cabo R, Zhao H, O'Brien CA, Manolagas SC, Almeida M, Sirtuin1 Suppresses Osteoclastogenesis by Deacetylating FoxOs, Mol Endocrinol. 29(10):1498– 509, 2015.

vii. Iyer S, Ambrogini E, Bartell SM, Han L, Roberson PK, Cabo R, Jilka RL, Weinstein RS, O'Brien CA, Manolagas SC, Almeida M, FoxOs attenuate bone formation by suppressing Wnt signaling, J Clin Invest. 123(8): 3404–3419, 2013.

viii. Iyer S, Han L, Bartel SM, Kim HN, Gubrij I, de Cabo R, O'Brien CA, Manolagas SC, Almeida M, Sirtuin 1 (Sirt1) Promotes Cortical Bone Formation by Preventing beta (β)-Catenin Sequestration by FoxO Transcription Factors in Osteoblast Progenitors, J Biol Chem. 289(35) :24069–24078, 2014.

These findings indicate that deacetylation of FoxOs by Sirt-1 may induce osteoblast differentiation in bone tissues.

According to the line which you suggest, we have added these findings and have revised the text in the revised manuscript (please find them in the revised manuscript, line 95-111). Thank you again for your comment.

  • More information on the three proteins in chondrocytes, why also chosen as cellular model?

Answer:

Thank you very much for your valuable comment. We understand what you mean: the reason why we have focused on the Glut-1 - Sirt-1 - Runx 2 axis in osteoblast and chondrocyte activities, especially mechanical stress-induced bone formation (osteogenesis). We are terribly sorry that our explanation was not enough.

       First of all, we took notice of Sirt-1 in osteoblasts and chondrocytes. As stated above, Sirt-1 is known to promote the osteoblast differentiation and osteogenesis, through transcription factor Runx 2 and/or FoxOs (as mentioned above). (However, it still unclear whether or not Sirt-1 may regulate the mechanical stress-mediated cellular responses such as osteogenesis. This is our main theme.) In addition, it has been demonstrated that Sirt-1 is required for promoting chondrogenic differentiation of mesenchimal stem cells (reference 52).

Also, it has been reported that Sirt-1 has important roles as a key regulator for cellular energy metabolism (energy sensor) as well as a master regulator of the mechanical stress response in a variety of cells including osteoblasts and chondrocytes.

These findings caught our attention. In the present study, we focused on the interaction of glucose transporter (Glut-1, which is expressed on the surface of cell membrane) with the energy sensor (Sirt-1). Since Sirt-1 play a role as an energy sensor of cellular energy metabolism (glucose uptake), it is known that the level of Glut-1 is inversely correlated with that of Sirt-1 in cell energy metabolism. We postulated that the Glut-1 on cell surface membrane may function as an extrinsic mechanical stress-sensor and may regulate the activities of Sirt-1- Runx2 in both osteoblasts and chondrocytes, since it has been already demonstrated that Sirt-1 inhibits the activity of Runx2, transcription factor of osteogenesis and chondrocyte hypertrophy (as above mentioned).

From the results of these findings, we speculated that that the Glut1-SIRT1-Runx2 pathway in osteoblasts and/or chondrocytes may play some sort of roles in mechanical stress-induced bone formation and osteoblast differentiation (revised Figure 7B = previous Fig.8).

 We agree that our explanation in the introduction was not enough. In the revised manuscript, we have revised the issue in the introduction section (line 95-100) and the discussion section (line 338-339).

We appreciate your helpful comment.

  • Some passages are repetitive, this could be shortened (e.g., line 113-116)

Answer:

Thank you for your helpful comment. As your suggestion, we have shortened repetitive passages and revised the “introduction” section without loss of our point in the revised manuscript (please find them in the revised manuscript).

Thank you again for your comment.

Material and Methods

  • Please indicate in the figure legends or somewhere else where parametric and where non-parametric statistical tests were used

Answer:

Thank you for your helpful comment. We are terribly sorry for lack of the number of test samples in each experiment. In the present study, 12 collagen sponge samples per each subgroup (control group, mechanically loading group, Glu-1 inhibitor-treated group: each group/each time point, n= 12) were used in one experiment. We used one 12-well culture dish for 12 samples/subgroup/each time point per one experiment of osteoblasts or chondrocytes. Each experiment was triplicated (12 individual samples/subgroup/each time point x 3 experiments). Basically, a one-way analysis of variance (ANOVA) with Bonferroni multiple comparison post-hoc test was used as parametric data sets in the present study according to statistician.

         As your suggestion, we have indicated the issue in the revised methods section. Thank you again for your comment.

  • Where the used chondrocytes all from the same OA patient? Could the “OA character” of the cells influence the results?

Answer:

Thank you for your valuable comment. In the present study, we isolated chondrocytes from only one OA patient who underwent the arthroplastic surgery. We got enough amount of chondrocytes from the relatively normal parts of cartilage explants. In contrast, we could not obtain useful chondrocytes from osteoarthritic degenerated cartilage explants. Then, we used them as a relatively normal chondrocytes in this study. Regarding the experiments with osteoblasts, we used a commercially available osteoblast strain. We are wondering if there may be probably little influence of the disease activity and OA character in the results of chondrocytes experiments. However, we understand that further studies are needed to clarify the influence of OA character of chondrocytes from patient, we have mentioned the issue in the revised manuscript (line 398-399):

[Enough number of cultured chondrocytes were obtained from the relatively normal parts of surgically resected cartilage tissues.]

Thank you again for your comment.

Results

  • The 3 of 3D in the title of the first subsection has moved to the subsection numbering (“2.1.3. D cell−“)

Answer:

Thank you for your helpful comment. We have revised the title “3D-collagen scaffold ~” in the revised manuscript (line 811) (previous "2.1. ~" moves to "4.3. ~" in methods section of revised manuscript, according to the reviewer's comment).

Thank you again for your comment.

  • Line 134: test should be tested

Answer:

We are awfully sorry for our careless mistake. We have revied “tested 3D” in the revised manuscript (line 417). Thank you for your helpful comment.

  • Figure 2: How was the quantification of microgranules on SEM Images done?

Answer:

Thank you very much for your valuable comments. As your comment, we had already done the quantitative analysis of microgranules on the cell surface on SEM images. In each subgroup of osteoblast study or chondrocyte study.

         [The number of microgranules per 10 cells/one collagen sponge sample (n= 12 samples/subgroup/each time point in a 12-well culture dish) were counted by three independent researchers. The analysis was triplicated (12 individual samples/subgroup/each time point x 3 experiments), and then the mean number of microgranules per one cell surface were calculated.] We added these sentences in the methods section (line 848-851).

         [The results clearly indicated that the mechanical force significantly induced the expression (number/cells) of cell surface microgranules in both osteoblasts [(24 hour-incubation group: mechanical loading (-) vs. mechanical loading (+) = 24.3 +/- 9.5/cells vs. 81.1 +/- 30.6/cells, p< 0.01), 48 hour-incubation group: mechanical loading (-) vs. mechanical loading (+) = 31.4 +/- 10.6/cells vs. 115.8 +/- 43.6/cells), p< 0.01] and chondrocytes [(24 hour-incubation group: mechanical loading (-) vs. mechanical loading (+) = 23.3 +/- 9.5/cells vs. 76.4 +/- 28.4/cells, p< 0.01), 48 hour-incubation group: mechanical loading (-) vs. mechanical loading (+) = 43.7 +/- 14.5/cells vs. 94.1 +/- 33.9/cells), p< 0.01].]

        In the present study, we have added the results in the revised version (results section, line 147-155). We appreciate your helpful comment.

  • Was the scaffold structure affected by the mechanical loading?

Answer:

Thank you for pointing that out. First of all, in the pilot experiments, we checked whether or not the mechanical force, which was tested in the present study, affected the structure of collagen sponge scaffold with no cell culture.

[The level of mechanical loading studied in the present study did not induce the destruction of collagen sponge. After removing the weight (mechanical loading), the compressed collagen sponges were restored.]

         We have added the issue as mentioned above in the revised manuscript (methods section, line 418-420).

  • Please add an SEM-image of the 48h non-loaded control (can also be in the supplementary).

Answer:

Thank you for your comment. We are sorry; we had omitted the 48h non-loaded controls, since no significant difference was observed between the 24h and 48h no-loading controls. As your comment, we have added the SEM-images of the 48 h non-loaded controls of both osteoblast group and chondrocyte group as the revied Figure 1C and 1G, respectively.

  • Figure 3: There is an error in the y-axis labeling of osteocalcin (“osteocalxin”).

Answer:

Thank you for your helpful comment. It was our fault. We have revised the Y-axis labeling of osteocalcin in the revised Figure 2 (previous Figure 3).

  • Figure 4: What happens when chondrocytes where incubated with mechanical loading for 48h?

Answer:

We are sorry for lacking the data of 48 h-mechanical loaded chondrocytes. We have added the data (levels of production of proteoglycan and type II collagen by chondrocytes) of 48 h-loaded chondrocyte groups in the revised Figure 4. Indeed, no significant differences in these two factors were observed between the 24 h-loaded chondrocytes group and the 48 h-loaded chondrocytes group.

In the revised manuscript, we have revised the revised Figure 3 and the figure legends (previous Fig.4). Thank you again for your comment.

  • Figure 5: Please explain why GAPDH was chosen as housekeeping gene. Normalization of glucose dependent proteins on GAPDH is normally considered difficult.

Answer:

Thank you for pointing that out. During the beginning of the present study, we had done the western blotting analysis using GAPDH. Indeed, we could sometimes get the data and at other time not. The results often varied even if we used same samples. Then, we changed “GAPDH” to “β-actin”. As you comment, we agree that normalization of glucose dependent proteins on GAPDH is normally considered difficult.

       In the present study, since the middle of study duration, we used “β-actin” in the western blotting. We are awfully sorry; it was our fault. We took the labels of GAPDH wrong with those of β-actin in the revised Figure 4 and 5 (previous Fig. 5 and 6).

  In the revised manuscript, we have revised those labels of “β-actin” in the revised Figure 5 and 6.

   Thank you again for your helpful comment.

  • Figure 6 = twice

Answer:

We are terribly sorry. It was our fault. We have revised the issue in the revised manuscript.

  • Figure 6: It seems like there is also for chondrocytes a tendency for increased Runx2 with loading. Please mention this in the text.

Answer:

Thank you very much for your helpful comment. We agree with your comment. Indeed, there were for chondrocytes tendencies for increased Sirt-1 and Runx2 with mechanical loading, although no significant differences were observed between the mechanically loaded chondrocytes group and the non-loaded chondrocytes group (Previous Fig. 6 = revised Fig. 5: P> 0.05, n= 12 collagen sponge samples per subgroup/each time point/one experiment, each experiment was triplicated).

       In the revised manuscript, we have mentioned the issue in the text (line 214-217). Thank you again for your comment.

  • 5: Please add the role of WZB117 (Glut-1 inhibitor)

Answer:

Thank you for your comment. As you suggest, we have added the information of Glut-1 inhibitor agent “WZB117” in the revised manuscript (line 479-482): 

[It has been reported that WZB117 has a potential to inhibit the expression of Glut-1 on cell surface membrane. Previous reports demonstrated that approximately 10 microM order of WZB117 seems to be suitable for the Glut-1 inhibition assay with Glut-1 inhibitor. We also pre-studied the suitable concentration of WZB117 in the pilot study and have chosen 10 microM of WZB117 for Glut-1 inhibition analysis.]

    In the revised manuscript, we have mentioned the issue (methods section, line 479-482).

  • Figure 7: One of the two controls could get into the supplementary, two controls are rather confusing.

Answer:

We appreciate your comment. We agree with your comment.

In previous Figure 7 (revised Fig.6), control 1 means the "incubation with medium only" group, and control 2 means "incubation with DMSO + medium" group. Glut-1 inhibitor, WZB117, is poorly water-soluble substance. Thus, in the present study, we dissolved WZB117 in DMSO solution. Therefore, DMSO solution was used as a control to WZB117-treated group, to checke whether or not DMSO itself influenced the expression of Glut-1, Sirt-1 and Runx2.

         We are sorry; two controls are rather confusing. If possible, we would like to change "Control 1"group to "medium only"group, and "control 2" to "control (DMSO medium)"group in the revised Figure 6.

Indeed, the results of medium only group in revised Figure 6A (expression of Glut-1), B (Sirt-1), C (Runx2) correspond to those of Figure 4A (Glut-1), B (Sirt-1), C (Runx2).

If possible, we may integrate the results of "medium only group" in revised Figure 6 into those of revised Figure 4. Both sample numbers for results in revised Figure 4 and Figure 6 were equally 12 samples/subgroup. We may revise and integrate the results of the revised Figure 4 and Figure 6 without loss of our point.

According to your comment, we have revised the Figure 4 and 6, text and the figure legend about the controls in the revised version.

       Thank you very much for your valuable comment.

  • Figure 7: Why was here normalized to β-Actin levels? Is β-Actin a good housekeeper when working with mechanical loading.

Answer:

Thank you for pointing that out to us. As we mentioned above, during the beginning of the present study, we had done the western blotting analysis using GAPDH. We could sometimes get the data and at other time not. The results often varied even if we used same samples. Then, we changed “GAPDH” to “β-actin”. As you comment above, we agree that normalization of glucose dependent proteins on GAPDH is normally considered difficult.

       In the present study, since the middle of study duration, we used “β-actin” in the western blotting. As a result, stable data were obtained in the western blotting. We have revised all Figures in the revised version. Thank you again for your helpful comment.

  • Does incubation with Glut-1 inhibitor also influence AP and osteocalcin levels?

Answer:

We appreciate your comment. As you suggest, in the present study, we have already found that the mechanical loading-induced upregulations of AP and osteocalcin levels were not observed in the Glut-1 inhibitor (+) groups. This means that mechanical loading could not induce the osteoblast activity in the presence of Glut-1 inhibitor. In addition, no significant differences in the AP and osteocalcin levels were observed between the non-loading controls and Glut-1 inhibitor (+) groups even in the presence or absence of mechanical loading.

       In our opinion, we think that mechanical loading induces the level of AP and osteocalcin through the mechanism of acceleration of “osteogenic transcription factor, Runx2" level in osteoblasts (the revised Figure 4EF, Figure 6C). As shown in the revised Figure 6C, since the Glut-1 inhibitor inhibits the loading-accelerated expression of Runx2, as a result, we think that the Glut-1 inhibitor, WZB117, may inhibit the levels of ALP and osteocalcin in osteoblasts.

       If possible, we would like to submit the results of ALP and osteocalcin as a supplementary data (please find them in the attached supplementary data file).

         Thank you very much for your helpful comment.

  • Figure 5+6+7: Please explain in the figure legend which exemplary bands of the blots are shown and what the frames mean. Additionally, please provide the full western blot images.

Answer:

Thank you for pointing that out to us. We agree with your comment. We carefully checked exemplary bands of western blots and marked then with frames (not to mistake them for something else).

According to your comment, in the revised manuscript, we have revised the explanation of representative exemplary bands in the full western blot images (revised Figure 4,5,6).

As we mentioned above, indeed, the results of "medium only" group in the revised Figure 6A (expression of Glut-1), B (Sirt-1), C (Runx2) correspond to those of the revised Figure 4A (Glut-1), B (Sirt-1), C (Runx2). If possible, we may integrate the results of "medium only" group in the revised Figure 6 into those of Figure 4. Both sample numbers for results in Figure 4 and Figure 6 were equally 12 samples/subgroup. We may revise and integrate the results of the revised Figure 4 and Figure 6 without loss of our point.

   Thank you again for your comment.

  • What about AMPK levels? It is proposed that the connection between Glut-1 and Sirt-1 can be mediated by AMPK.

Answer:

Thank you very much for your valuable comment. We agree that your comment is very important to clarify the exact mechanism of mechanical stress-induced osteogenesis (bone formation) via the Glut-1 - Sirt-1 - Runx2 pathway.

       As you comment, AMPK is well known to be an energy sensor as well as Sirt-1 in cellular energy metabolism (ATP production). Numerous reports have already demonstrated that AMPK level is positively correlated with the level of Sirt-1 in a variety of cells (Discussion section). In our previous studies, we also observed the correlation between the levels of Sirt-1 and AMPK (references 31). Sirt-1 is known to crosstalk with AMPK, forming a SIRT1-AMPK positive feedback loop, in cellular energy metabolism (reference 23,50,51) (we have already discussed the issue in the Discussion section).

       Indeed, in the pilot study, we also observed that the loading decreased the expression of AMPK as well as Sirt-1 in osteoblasts. From the results of previous studies, we postulate that the mechanical loading may induce the increased level of Glut-1 and its resultant decreases of both energy sensors, Sirt-1 and AMPK. As shown in the revised Figure 7, main purpose of this study was to clarify the mechanism of the loading-induced bone formation (osteogenesis), with a sclerostin-Wnt/beta-catenin-independent signaling pathway. Since it is known that Sirt-1 (but not AMPK) negatively regulates the level of Runx2 (a master transcription factor for osteogenic and chondrogenic differentiation, Chondrocyte hypertrophy), we focused on Sirt-1 among Sirt-1 and AMPK.

In the revised manuscript, we have mentioned the interaction of Sirt-1 with Runx 2 as following in the discussion section (line 316-323) :

      [From the results of previous studies, we postulate that the mechanical loading may induce the increased level of Glut-1 and its resultant decreases of both energy sensors, SIRT1 and AMPK. As shown in the Figure 7, main purpose of this study was to clarify the mechanism of the loading-induced bone formation (osteogenesis), with a sclerostin-Wnt/beta-catenin-independent signaling pathway. Since it is known that SIRT1, but not AMPK, negatively regulates the level of Runx2 (a master transcription factor for chondrocyte hypertrophy, osteogenic and chondrogenic differentiation), we focused on SIRT1 among SIRT1 and AMPK.]

Thank you again for your valuable comment.

  • The β in β-Actin is replaced by some error sign in the manuscript (e.g., figure 7 figure legend).

Answer:

We are terribly sorry for some error sign “β” and “µM”. It seems that there is something wrong with "font system" in our PC. In the revied manuscript, we have carefully checked and revised all error sign of “β” and “µM”.

         Thank you very much for your comment.

Discussion

  • Some repetitive passages (line 325-327; 341-345)

Answer:

Thank you for your helpful comment. We agree with your comment. We have revised some repetitive passages in all sections as well as discussion section, without loss of our point (please find them in the revised manuscript). Thank you for your comment.

  • What happens with cyclic mechanical stress or longer incubation with the weight?

Answer:

Thank you very much for your valuable comment. We are grateful for your suggestion. We have also considered the difference in the mechanical stress response among the continuous/physiologic loading, the cyclic loading and the long-term loading.

         The purpose of our study is to clarify the mechanism of mechanical loading-induced bone formation (osteoblast activity) with a sclerostin/wnt-independent pathway (Figure 7). Of course, we understand that we have to study the mechanism of cellular responses under the condition of continuous/physiologic stress and the cyclic stress.

         First of all, in the present study, we examined the cellular response under the condition of continuous/physiologic stress in osteoblasts and chondrocytes. (In our opinion, over loading may induce the apoptosis or down-regulation of osteoblastic activity, consequently resulting in the decrease of bone formation.)

         Nextly, based on the results of the present study, we plan to study the osteoblast activity in response to the cyclic stress using a 3D-cell culture system. We will do the comparative study on the condition of continuous/physiologic loading, the cyclic loading and the long-term loading. We are now preparing a newly developed cyclic loading machine.

         In near future, we will show the different mechanism in the mechanical loading-induced osteogenesis between the cyclic loading and continuous loading conditions.

         According to your suggestion, we will expand our study. In the revised manuscript, we have discusses the issue as mentioned above (Discussion section, line 356-359):

         [We have also considered the difference in the mechanical stress response among the continuous/physiologic loading, the cyclic loading and the long-term loading. we plan to study the osteoblast activity in response to the cyclic stress using a newly developed cyclic loading 3D-cell culture system.]

         Thank you again for your valuable comment.

  • Please discuss more why mechanical loading does not induce chondrogenic differentiation. This is contrary to the common sense.

Answer:

Thank you very much for your comment. As you comment, it has been demonstrated that the mechanical loading induces chondrocyte differentiation and proliferation in vitro. In contrast, it is well known that obesity and excessive mechanical loading induces the degeneration of articular cartilage and the downregulation of chondrocyte activity, as an OA-inducible factor. To maintain the function of articular cartilage in the joint, chondrocytes might have a mechanical stress tolerance to some degree.

       Indeed, articular cartilage is avascular tissue with little or no potential of regeneration. Articular cartilage is a highly organized tissue composed primarily of proteoglycan, type II collagen and hyaluronic acid with a small amount of other proteins including elastin, type IX and type X collagen, which has poor spontaneous self-healing capacity. In contrast, bone tissue with rich vascular system has a potential of remodeling with high potential of regeneration. It is well known that avascular cartilage tissue shows almost no potential of regeneration and adult chondrocytes into the tissue have little or no potential of proliferation (but in vitro cultured chondrocytes have a potential to proliferate).

         Although further studies are needed to verify the difference in the mechanical stress response between osteoblasts (bone tissue) and chondrocytes (articular cartilage), there might be some different mechanism in the stress tolerance between the two.

         Your comment regarding the difference between the two are critical important to expand our study of the stress response to mechanical loading.

         In the revised manuscript, we have discussed the above mentioned issues (line 361-368) :  (~ In the present study, mechanical loading did not induce chondrocyte activity.) Articular cartilage is an avascular tissue, which has poor spontaneous self-healing capacity. Adult chondrocytes into the cartilage tissue have little or no potential of proliferation. In contrast, bone tissue with rich vascular system has a potential of remodeling with high level of regeneration. To maintain the function of articular cartilage in the joint, chondrocytes might have a mechanical stress tolerance to some degree. Although further studies are needed to verify the difference in the mechanical stress response between osteoblasts and chondrocytes, there might be some different mechanism in the stress tolerance between the two.]

              Thank you very much for your valuable review. I would very much appreciate reviewing our revised manuscript and answer letter.

Reviewer 2 Report

The present study reports on findings regarding the involved mechanisms of the direct response of osteoblasts to mechanical stress.

The abstract is clear; however, it is a bit too elaborate.

An introduction is well structured. The background information is clear, the significance of the research results is justified.

Subsection “2.1.3. D cell−collagen scaffold constructs for human cultured cells.” Belong to the materials and methods section as the given information describes the experimental design rather than the result.

Moreover, the number of test samples should be clearly stated as for now it is not clear. “After incubating overnight, one round 3D collagen scaffold disc was placed into each well of a 12-well culture dish.” One round sample was subsequently placed in each of the 12 wells or did you used 12 samples that were placed in all of 12 wells of a 12-well culture dish?

Figure captions are too complex and mainly repeating the information already known from the text. That should be reworked. The figure captions should be more concise.

In section “2.2. Effects of mechanical loading on cellular morphology in 3D cell culture.” The authors show the SEM images and discuss the increasing amount of “microgranlues”. There is no further discussion of the significance of “microgranules” and why their amount increased upon loading. Are those “microgranlues” calcium deposits? Probably Alizarin red staining could provide additional information and could also confirm the process of differentiating osteoblast cells to mature osteoblasts with mineralization of matrix. As long as you used SEM for cell morphology analysis an EDX of those deposits could be also performed to confirm the elemental composition of. “microgranlues”. Besides, the methodology of “microgranlues” counting is not stated. How many SEM images were obtained and how many measurements were taken, what is the statistical significance of those measurements?

I would like to comment on further results being reported. I would not recommend using bar graphs as a mean to report your experimental results in case if you are working with small sample data. Figures in scientific publications are critically important because they often show the data supporting key findings.

One cannot use descriptive statistics to analyze and report data having small sample numbers. Mean/SEM/SD are inappropriate in this context. I suggest authors review recent papers on this subject - e.g. Weissberber et al PLOS Biology 2015; Nuzzo Nature 2014. Appropriate statistical evaluation should be applied to all data. I would suggest using the Gardner–Altman the Cumming plot styles.

Statistical analysis should be appropriately elaborated in the experimental section and the number of samples and experiments clearly stated. It is not recommended to report results in such a manner: “The results of each experimental condition were determined from the mean of triplicate experiments” as it is not very clear for readers. If your study design/sample size meet established criteria to justify the use of descriptive statistics then please provide the information to readers so we can evaluate independently. This means one has to provide the following information: sample size determination, statistical power, size effect, multiple group comparison, randomization.

I would like to mention that poor statistical assessment is one of the major causes for low-quality results being published (https://elifesciences.org/articles/48175).

The appropriate evaluation of obtained experimental results will strengthen the manuscript and your key findings.

There are some small mistakes throughout the text of the manuscript, which are not critical. For example, in subsection “4.1. Monolayer human cell cultures: chondrocytes, osteoblasts, and osteoclasts.” Information concerning osteoclasts is not given. Perhaps it is redundant.

Overall, the reported findings are interesting, however, as for now, the manuscript should be modified in a way that the readers won’t have any questions about the reliability of the reported results.

Author Response

Answer letter to Reviewer 2

  • The abstract is clear; however, it is a bit too elaborate.

Answer:

Thank you for your helpful comment. As your suggestion, we have shortened repetitive passages and revised the “abstract” section without loss of our point in the revised manuscript. Thank you again for your comment.

  • An introduction is well structured. The background information is clear, the significance of the research results is justified.

Answer:

We are grateful for your comment.

  • Subsection “2.1. 3D cell−collagen scaffold constructs for human cultured cells.” Belong to the materials and methods section as the given information describes the experimental design rather than the result.

Answer:

Thank you very much for your helpful comment. We agree with your comment. According to your comment, we have revised "this subsection and the previous Fig. 1" move to "Materials and methods (line 412-422)" section in the revised manuscript (previous Fig.1 >>> revised Fig. 8). Thank you again for your comment.

  • Moreover, the number of test samples should be clearly stated as for now it is not clear. “After incubating overnight, one round 3D collagen scaffold disc was placed into each well of a 12-well culture dish.” One round sample was subsequently placed in each of the 12 wells or did you used 12 samples that were placed in all of 12 wells of a 12-well culture dish?

Answer:

Thank you very much for your helpful comment. We are terribly sorry for lack of the number of test samples in each experiment. In the present study, 12 collagen sponge samples per each subgroup (control group, mechanically loading group, Glu-1 inhibitor-treated group: each group/each time point, n= 12) were used in one experiment. We used one 12-well culture dish for 12 samples/subgroup/each time point per one experiment of osteoblasts or chondrocytes. Each experiment was triplicated (12 individual samples/subgroup/each time point x 3 experiments).

       In the revised manuscript, we have stated the number of samples/each subgroup in all experiments. Thank you again for your comment.

  • Figure captions are too complex and mainly repeating the information already known from the text. That should be reworked. The figure captions should be more concise.

Answer:

Thank you very much for your comment. We agree with your comment.

According to your comment, we have revised all figure captions more concisely in the revised manuscript.

  • In section “2.2. Effects of mechanical loading on cellular morphology in 3D cell culture.” The authors show the SEM images and discuss the increasing amount of “microgranlues”. There is no further discussion of the significance of “microgranules” and why their amount increased upon loading. Are those “microgranlues” calcium deposits? Probably Alizarin red staining could provide additional information and could also confirm the process of differentiating osteoblast cells to mature osteoblasts with mineralization of matrix. As long as you used SEM for cell morphology analysis an EDX of those deposits could be also performed to confirm the elemental composition of. “microgranlues”. Besides, the methodology of “microgranlues” counting is not stated. How many SEM images were obtained and how many measurements were taken, what is the statistical significance of those measurements?

Answer:

Thank you very much for your helpful comments.

We are terribly sorry; the analysis to confirm the elemental composition of microgranules has still uncompleted. According to your comments, we would like to perform the SEM for cell morphology analysis. Based on the result of the present study, we now plan to study the osteoblast activity in response to the cyclic stress using a 3D-cell culture system. We will also do the SEM analysis again on the condition of continuous/physiologic loading, the cyclic loading and the long-term loading. We are now preparing a newly developed cyclic loading machine. In near future, we will show the different mechanism in the mechanical loading-induced osteogenesis between the cyclic loading and continuous loading conditions through the SEM analysis.

       Regarding the quantitative analysis of microgranules, we had done the quantitative analysis of microgranules on the cell surface on SEM images. As above mentioned, in the present study, 12 collagen sponge samples per each subgroup (control group, mechanically loading group: each group/each time point, n= 12) were prepared in the SEM analysis. The analysis was triplicated (12 individual samples/subgroup/each time point x 3 experiments).

         In each subgroup of osteoblast study or chondrocyte study, the number of microgranules per 10 cells/one collagen sponge sample (n= 12 samples/subgroup/each time point) were counted by three independent researchers, and then the mean number of microgranules per one cell surface were calculated.

         The results clearly indicated that the mechanical force significantly induced the expression (number) of cell surface microgranules in both osteoblasts [(24 hour-incubation group: mechanical loading (-) vs. mechanical loading (+) = 24.3 +/- 9.5/cells vs. 81.1 +/- 30.6/cells, p< 0.01), 48 hour-incubation group: mechanical loading (-) vs. mechanical loading (+) = 31.4 +/- 10.6/cells vs. 115.8 +/- 43.6/cells), p< 0.01] and chondrocytes  [(24 hour-incubation group: mechanical loading (-) vs. mechanical loading (+) = 23.3 +/- 9.5/cells vs. 76.4 +/- 28.4/cells, p< 0.01), 48 hour-incubation group: mechanical loading (-) vs. mechanical loading (+) = 43.7 +/- 14.5 vs. 94.1 +/- 33.9), p< 0.01].

         In the present study, we have added the information of sample number and the results in the revised version (line 147-155). We appreciate your helpful comment.

  • I would like to comment on further results being reported. I would not recommend using bar graphs as a mean to report your experimental results in case if you are working with small sample data. Figures in scientific publications are critically important because they often show the data supporting key findings.

One cannot use descriptive statistics to analyze and report data having small sample numbers. Mean/SEM/SD are inappropriate in this context. I suggest authors review recent papers on this subject - e.g. Weissberber et al PLOS Biology 2015; Nuzzo Nature 2014. Appropriate statistical evaluation should be applied to all data. I would suggest using the Gardner–Altman the Cumming plot styles.

       Statistical analysis should be appropriately elaborated in the experimental section and the number of samples and experiments clearly stated. It is not recommended to report results in such a manner: “The results of each experimental condition were determined from the mean of triplicate experiments” as it is not very clear for readers. If your study design/sample size meet established criteria to justify the use of descriptive statistics then please provide the information to readers so we can evaluate independently. This means one has to provide the following information: sample size determination, statistical power, size effect, multiple group comparison, randomization.

       I would like to mention that poor statistical assessment is one of the major causes for low-quality results being published (https://elifesciences. org/articles/48175). The appropriate evaluation of obtained experimental results will strengthen the manuscript and your key findings.

Answer:

We are awfully sorry again for lack of the number of test samples in each experiment.

In the present study, 12 cell-collagen sponge samples per each subgroup (control group, mechanically loading group, Glu-1 inhibitor-treated group: each subgroup/each time point, n= 12) were used in one experiment. We used one 12-well culture dish for 12 samples/subgroup/each time point per one experiment of osteoblasts or chondrocytes. Each experiment was triplicated (12 individual samples/subgroup/each time point x 3 experiments).

Regarding the sample number for western blotting and ELISA analyses, although our individual experiments with 12 samples per subgroup/each time point might be inferior to the other papers published in the journal, we would very much appreciate reviewing our revised manuscript.

       In the revised manuscript, we have stated the number of samples/each subgroup in all experiments. Thank you again for your comment.

  • There are some small mistakes throughout the text of the manuscript, which are not critical. For example, in subsection “4.1. Monolayer human cell cultures: chondrocytes, osteoblasts, and osteoclasts.” Information concerning osteoclasts is not given. Perhaps it is redundant.

Answer:

Thank you very much for your helpful comment. We are so sorry for our careless mistake. We have carefully checked and have revised some mistakes ("osteoclasts", some error signs “β” and “µM”, etc.)

       Thank you again for your helpful comment.

  • Overall, the reported findings are interesting, however, as for now, the manuscript should be modified in a way that the readers won’t have any questions about the reliability of the reported results.

Answer:

We appreciate your comment. We agree with your comments.

According to the line which all reviewers suggest, we have carefully revised text and figures (careless mistakes, statements of sample numbers, statistical analysis, supplemental data, quantitative analysis of microgranules in SEM images, full images of western blotting, ELISA data ) in the revised manuscript.

We would very much appreciate reviewing our revised manuscript and answer letter.

Round 2

Reviewer 1 Report

A far as I can read the PDF (somehow the format was corrupted - some lines are double etc.) and the authros response all questions raised have been answered adequately. Althought he introduction is now even longer.

Author Response

Reviewer 1 - round 2

A far as I can read the PDF (somehow the format was corrupted - some lines are double etc.) and the authors response all questions raised have been answered adequately. Although the introduction is now even longer.

Answer:

Thank you very much for your review and comment.

According to your comment, we have shortened our revised manuscript (round 2) to about 70% of the previous version.

We have checked the text, again. We described all necessary information in the main text. And then, some repetitive & redundant passages were removed from the figure legends.

Thank you again for your valuable review. We would appreciate it very much if you could accept our revision.

Reviewer 2 Report

"Thank you very much for your helpful comment. We agree with your comment. According to your comment, we have revised "this subsection and the previous Fig. 1" move to "Materials and methods (line 412-422)" section in the revised manuscript (previous Fig.1 >>> revised Fig. 8). Thank you again for your comment."

Please be aware that all figures and tables should appear in the order of their numbers as well as their first mention in the text as close as possible to their first mention.

"Thank you very much for your comment. We agree with your comment.

According to your comment, we have revised all figure captions more concisely in the revised manuscript."

I did not notice the change. Moreover, in the revised version more information has been added. In my opinion, long captions are hard to comprehend. For example in the caption for figure 6 I see more than twenty lines of text which is too much. Please describe all necessary information in the main text.

Although some corrections in the section "Effects of mechanical loading on cellular morphology in 3D cell culture" were made, unfortunately, co-authors did not improve the discussion. Please describe the significance of "microgranules" and what are they in more detail, ideally with some reference to existing literature.

The plot styles were not modified. Please consider using the Gardner–Altman or the Cumming plot styles as was mentioned in my previous review. Please revise the following papers (10.1038/s41592-019-0470-3), (10.1371/journal.pbio.1002128).

I did not find in the text how you tested the normality of your data distribution. Moreover, in the bar graphs, the information regarding the individual data points is hidden from the reader. Were there outliers in your data?

Unfortunately, you did not report the statistical power or size effect for your studies. On the other hand, the limitations of your study became more obvious for the reader after you reported the number of samples in subgroups and replicates for each experiment more carefully.

Overall, the authors made an effort to improve the quality of the text.  However, the manuscript could benefit from a more detailed statistical analysis and improved data visualization. 

Author Response

Reviewer 2 - round 2

Our previous Answer:

Thank you very much for your helpful comment. We agree with your comment. According to your comment, we have revised "this subsection and the previous Fig. 1" move to "Materials and methods (line 412-422)" section in the revised manuscript (previous Fig.1 >>> revised Fig. 8). Thank you again for your comment."

 Please be aware that all figures and tables should appear in the order of their numbers as well as their first mention in the text as close as possible to their first mention.

Answer:

Thank you very much for your comment.

In the first submitted manuscript, the figure for "3D cell-collagen scaffold constructs" was placed in the "results section", since we thought that this was the first appearance and mention in the text: the reason why the IJMS journal is constructed in order of "Introduction", "Results", "Discussion", and the last "Materials and Methods" sections (but not, in order of "Introduction", "Materials & Methods", "Results", "Discussion").  (In the first submission, we had described the information of 3D cell-collagen scaffold in the second section "Results", but not in the last section "Materials and Methods", as the first appearance and first mention.)

              We are awfully sorry for our mistake. We misunderstood your previous comment (Round 1);  in spite of the first appearance and mention of the 3D collagen construct, we had mistakenly moved "the previous Fig. 1" to the last section "Materials and methods" with the experimental designs in the revised (round 1) manuscript (previous Fig.1 moved to the revised Fig. 8).

              How should we revise the issue? If you do not mind, we would like to put "previously revised Fig. 8" back in the "Result section" (as a new Fig. 1: as the first appearance/mention in the text), but we would like to remain the "description of experimental designs (materials & methods)" in the last "Methods section".

We would appreciate it very much if you could accept our revision.

Our previous Answer:

Thank you very much for your comment. We agree with your comment.

According to your comment, we have revised all figure captions more concisely in the revised manuscript."

I did not notice the change. Moreover, in the revised version more information has been added. In my opinion, long captions are hard to comprehend. For example, in the caption for figure 6 I see more than twenty lines of text which is too much. Please describe all necessary information in the main text.

Answer:

Thank you very much for your comment. We are terribly sorry; we misunderstood your comment "too long figure captions" to "long subtitles of figure legends".

              We have shortened and revised all figure captions in new revised manuscript. As your comment, we described all necessary information in the main text. And then, some repetitive passages were removed from the figure legends.  Thank you again for your comments.

Although some corrections in the section "Effects of mechanical loading on cellular morphology in 3D cell culture" were made, unfortunately, co-authors did not improve the discussion. Please describe the significance of "microgranules" and what are they in more detail, ideally with some reference to existing literature.

Answer:

Thank you very much for your comment. We are sorry for not explaining sufficiently about "microgranules". In the revised version (round 2), we have discussed the issue in the discussion section as following:

[As shown in Figure 2B,D, continuous and physiologic mechanical loading induced the deposition of microgranules on the surface of collagen sponge fibers as well as on the cell surface of osteoblasts. The levels of microgranule deposition increased with incubation time on both surface of osteoblasts and collagen fibers, suggesting the production of calcium deposits and calcium mineralization by osteoblasts. However, the elemental composition of microgranules still remains unknown. To confirm the composition of microgranules, we plan to analyze the composition of microgranules and calcium mineralization by alizarin red staining. We have also considered the difference in the mechanical stress response among the continuous/physiologic loading, the cyclic loading and the long-term loading. Chen XI et al. demonstrated that cyclic compression load promoted osteoblast differentiation and maturation, leading to calcium mineralization/bone formation [49]. We plan to study the osteoblast activity and the calcium mineralization in response to the cyclic stress using a newly developed cyclic loading 3D-cell culture system.]

       We would appreciate it very much if you could accept our revision.

The plot styles were not modified. Please consider using the Gardner–Altman or the Cumming plot styles as was mentioned in my previous review. Please revise the following papers (10.1038/s41592-019-0470-3), (10.1371/journal.pbio.1002128).

I did not find in the text how you tested the normality of your data distribution. Moreover, in the bar graphs, the information regarding the individual data points is hidden from the reader. Were there outliers in your data?

Unfortunately, you did not report the statistical power or size effect for your studies. On the other hand, the limitations of your study became more obvious for the reader after you reported the number of samples in subgroups and replicates for each experiment more carefully.

Overall, the authors made an effort to improve the quality of the text.  However, the manuscript could benefit from a more detailed statistical analysis and improved data visualization. 

Answer:

We are terribly sorry; we misunderstood your valuable comments. It was our faults.

We should have been aware of your comments. We have checked the data from all experiments: the ELISA data (concentrations of parameters for osteoblast and chondrocyte activities) and western blotting analysis.

              As shown in Figure 3 (osteoblast activity: ALP, osteocalcin concentrations) and Figure 4 (chondrocyte activity: proteoglycan, type II collagen concentrations), we have revised each graph using the Gardener-Altman plot style:

Here, Gardener-Altman plot graphs (Please see in the PDF files).

Cohen's d between Control and mechanical loading group is shown in the above Gardner-Altman estimation plot. Each group is plotted on the left axes; the mean difference is plotted on a floating axes on the right as a bootstrap sampling distribution. The mean difference is depicted as a dot; the 95% confidence interval is indicated by the ends of the vertical error bar. The effect sizes and CIs are reported above as: effect size [CI width lower bound; upper bound].

For example of the results,

[ Regarding the level of ALP production from osteoblasts, the unpaired Cohen's d between Control and the loading (+)-24-hour incubation group was 0.313 [95.0%CI -0.599, 1.12] and the P value of the two-sided permutation t-test is 0.456 (Figure 3, A-a).  The unpaired Cohen's d between Control and the loading (+)-48-hour incubation group was 1.1 [95.0%CI 0.198, 1.87] and the P value of the two-sided permutation t-test is 0.0124 (A-b).]

We are sorry; our previous answer to your questions/comments (Round 1) may be misstatement.

In the present study, we have performed each experiment using a clonal normal osteoblastic cell strain (commercially available) or a clonal healthy chondrocyte culture from a patient with OA (normal parts of cartilage tissue, but not degenerated parts). Each subgroup/each time point consists of "n=12 collagen cell culture sponges". This means that 12 collagen-cell cultures (dishes)/subgroup were prepared from one same clonal cell strain, but not "12 independent samples from 12 different cell strains or from 12 patient samples".

              Since 12 cell cultures (3D cell-collagen sponge) are basically isolated from one same clonal cell culture, there may be probably little variation of cellular condition. Thus, in our opinion, in vitro ELISA assay, I wonder if 12 wells/subgroup might be generally suitable for one assay (in the one ELISA kit plate). However, we agree with your comment regarding the small number of samples. We have to consider the issue, and we have revised the bar graph to the Gardner-Altman estimation plot style.

              Indeed, we have again checked all references listed in our manuscript and many papers published in the IJMS journal, from the point of view of sample numbers and data plot styles for ELISA. Based on this observation, may it please your honour, we are wondering if 12 wells/subgroup from one cell culture stain may fulfill requirement for one assay. Also, the experiments (assays) were independently triplicated in all ELISA and western blot assays using normal osteoblasts and healthy chondrocytes.

              We think that the ELISA is a quantitative assay to measure the concentration of target agent in cell culture. Data from ELISA assay are absolute values.

              In contrast to the ELISA, we think that the western blot is a qualitative assay, but not a quantitative assay. Data of western blotting is based on the expression of target protein (bands on the western blotting gel membrane). Indeed, the level of expression of protein band is shown as a relative expression ratio to β-actin band (intensity of protein band is normalized with the respective β-actin band.) The ratio is a semiquantitative relative value, but not absolute value.

              Again, we have checked all data and plot styles of western blot experiments in all references listed in our manuscript and many papers published in the IJMS journal, from the point of view of sample numbers and data plot styles.

              We observed that most papers showed their data by only representative band images of western blot and/or bar graph styles of the semiquantitative relative values. Regarding the number of samples, most papers mentioned samples number: "n= 3~6, the data were obtained from three ~ four independent experiments" in their texts.

              May it please your hornour we are wondering if it is possible for the semiquantitative data from the western botting to be presented them by bar graphs in the IJMS journal.

We would appreciate it very much if you could accept our revision.
